# Dual Labeling of Primary Cells with Fluorescent Gadolinium Oxide Nanoparticles

**DOI:** 10.3390/nano13121869

**Published:** 2023-06-16

**Authors:** Nadine Brune, Benedikt Mues, Eva Miriam Buhl, Kai-Wolfgang Hintzen, Stefan Jockenhoevel, Christian G. Cornelissen, Ioana Slabu, Anja Lena Thiebes

**Affiliations:** 1Institute of Applied Medical Engineering, Helmholtz Institute, Medical Faculty, RWTH Aachen University, 52074 Aachen, Germany; nadine.brune@rwth-aachen.de (N.B.); jockenhoevel@ame.rwth-aachen.de (S.J.);; 2Institute of Pathology, Electron Microscopy Facility, University Clinic Aachen, 52074 Aachen, Germany; 3DWI—Leibniz-Institute for Interactive Materials, 52074 Aachen, Germany; 4Aachen-Maastricht Institute for Biobased Materials, Faculty of Science and Engineering, Maastricht University, 6167 RD Geleen, The Netherlands; 5Department of Pneumology and Internal Intensive Care Medicine, Medical Clinic V, University Clinic Aachen, 52074 Aachen, Germany

**Keywords:** rose bengal, dextran, contrast agent, magnetic resonance imaging (MRI), adipose-derived mesenchymal stromal cells (ASC)

## Abstract

The interest in mesenchymal stromal cells as a therapy option is increasing rapidly. To improve their implementation, location, and distribution, the properties of these must be investigated. Therefore, cells can be labeled with nanoparticles as a dual contrast agent for fluorescence and magnetic resonance imaging (MRI). In this study, a more efficient protocol for an easy synthesis of rose bengal–dextran-coated gadolinium oxide (Gd_2_O_3_-dex-RB) nanoparticles within only 4 h was established. Nanoparticles were characterized by zeta potential measurements, photometric measurements, fluorescence and transmission electron microscopy, and MRI. In vitro cell experiments with SK-MEL-28 and primary adipose-derived mesenchymal stromal cells (ASC), nanoparticle internalization, fluorescence and MRI properties, and cell proliferation were performed. The synthesis of Gd_2_O_3_-dex-RB nanoparticles was successful, and they were proven to show adequate signaling in fluorescence microscopy and MRI. Nanoparticles were internalized into SK-MEL-28 and ASC via endocytosis. Labeled cells showed sufficient fluorescence and MRI signal. Labeling concentrations of up to 4 mM and 8 mM for ASC and SK-MEL-28, respectively, did not interfere with cell viability and proliferation. Gd_2_O_3_-dex-RB nanoparticles are a feasible contrast agent to track cells via fluorescence microscopy and MRI. Fluorescence microscopy is a suitable method to track cells in in vitro experiments with smaller samples.

## 1. Introduction

Using mesenchymal stromal cells as a treatment option for lung diseases is a promising approach. It was previously shown in experimental trials that diseases such as chronic obstructive pulmonary disease, acute respiratory distress syndrome, and idiopathic pulmonary fibrosis responded well to this treatment option [1,2]. Further application options for mesenchymal stromal cell therapies arise in several different fields such as neurological or cardiovascular diseases [3,4]. Along with the rapidly increasing interest in cell therapy, the urge of locating these cells after application is one focus of interest [2,5].

A method for cell tracking should be noninvasive, have a high sensitivity, be non-toxic, and be biocompatible [6]. In addition, it is favorable to be able to track the cells with two or more complementary methods to gain more information and assure accuracy. Magnetic resonance and optical fluorescence imaging are both noninvasive and well-established diagnostic tools in clinical use. Fluorescence imaging has a high sensitivity to locate single cells but is limited in penetration depth. For assessments of deeper compartments, a second complementary method is needed. Magnetic resonance imaging (MRI) is able to detect signals in deeper areas of the body but is dependent on the local concentration of cells to get sufficient signals [6].

Thus, the main objective of this study was to develop a labeling method for cell detection in fluorescence microscopy and MRI. The used contrast agent should be easy to synthesize and not interfere with cell viability.

Gadolinium is known as an MRI contrast agent and is widely used in clinical examinations. Cell tracking in MRI via internalized gadolinium oxide nanoparticles (Gd_2_O_3_ nanoparticles) is described in several studies [6,7,8,9,10]. Kumar et al. prepared Gd_2_O_3_ nanoparticles covered with dextran (dex) which carries a rose bengal (RB) dye [10]. Dextran is a high molecular weight polysaccharide that is used to increase the biocompatibility and non-toxicity of the nanoparticles [11,12]. Rose bengal is established as a dye in ophthalmology. Here, it is used to detect damage to the epithelium of the ocular surface [13]. In the modified nanoparticles, rose bengal works as a fluorescent agent for optical tracking. Kumar et al. showed the nanoparticles to be efficient for both fluorescence and magnetic resonance imaging applications as well as cellular uptake in cancerous (A-549, U-87) and normal (HEK-293) cell lines [10].

However, the synthesis and modification of the Gd_2_O_3_-dex-RB nanoparticles described by Kumar et al. is very complex, involves toxic substances, and takes three days before the nanoparticles are ready to use [10]. In this study, we restructured and shortened the synthesis steps to one day and replaced the toxic substances for faster, easier, and safer preparation of rose bengal–dextran-coated Gd_2_O_3_ nanoparticles.

Moreover, adequate in vitro data on different cell lines and especially primary cells is lacking. Thus, the influence of Gd_2_O_3_-dex-RB nanoparticles on mesenchymal stromal cells was evaluated. The Gd_2_O_3_-dex-RB nanoparticles produced with the new protocol were characterized by zeta potential measurements, photometric measurements, fluorescence microscopy, magnetic resonance, and electron microscopy imaging. In in vitro cell experiments with a cell line (SK-MEL-28) and with primary adipose-derived mesenchymal stromal cells (ASC), nanoparticle internalization, and cell behavior were assessed.

## 2. Materials and Methods

### 2.1. Synthesis and Sample Preparations

#### 2.1.1. Preparation of Gd_2_O_3_ Nanoparticles

An amount of 10 mL of a 10 mM nanoparticle dispersion in ultrapure water (UPW) using gadolinium(III)-oxide nanopowder with the size of 20–40 nm (Alfa Aesar by Thermo Fisher Scientific, Kandel, Germany) was prepared, vortexed, and dispersed in an ultrasound bath (USC300D, VWR) for 10 min at room temperature (RT) (25 °C). UPW had a pH of 5.95. Dispersion was optimized in a pH series, which can be seen in Appendix A. For coating steps, dispersion was always stirred magnetically at 1400 rpm at RT (Figure 1).

#### 2.1.2. Synthesis of Gd_2_O_3_-dex Nanoparticles

Next, 1 mL of 6% 75-kD-dextran (Alfa Aesar) in UPW was prepared and added dropwise to the nanoparticle dispersion, which was then stirred for 1 h. The nanoparticle dispersion was transferred into centrifugation tubes (Eppendorf, Hamburg, Germany) to separate the nanoparticles from the water by centrifugation at 20,238× *g* for 7.5 min at RT. The supernatant was removed and 10 mL UPW was added. The nanoparticle pellet was vortexed, dispersed in an ultrasound bath for 3 min, and stirred for the next coating step (Figure 1).

#### 2.1.3. Synthesis of Gd_2_O_3_-dex-RB Nanoparticles

Then, 100 µL of 3% rose bengal (Alfa Aesar) in UPW was prepared and added dropwise to the nanoparticle dispersion. Protected from light, it was stirred for 1 h. Nanoparticles were separated from water by centrifugation. The supernatant was removed, and the nanoparticle pellet was dispersed in either 10 mL UPW for nanoparticle characterization assessments or in 10 mL cell culture medium (see Section 2.2) for MRI assessments and cell labeling (Figure 1).

### 2.2. Cell Isolation and Culture

#### 2.2.1. SK-MEL-28

Melanoma cells of the human cell line (SK-MEL-28) were purchased from CLS GmbH, Eppelheim, Germany. Cells were expanded using Dulbecco’s Modified Eagle Medium + GlutaMAX-I (DMEM, Thermo Fisher Scientific, Kandel, Germany) with 1% antibiotic-antimycotic solution (ABM, Thermo Fisher Scientific) and 10% Fetal Calf Serum (FCS, Thermo Fisher Scientific), incubated at 37 °C and 5% CO_2_. Culture medium was changed every two days. At 70–80% confluency, cells were washed with phosphate-buffered saline (PBS, Thermo Fisher Scientific) and passaged using Trypsin/EDTA (PAN-Biotech, Aidenbach, Germany).

#### 2.2.2. Adipose-Derived Mesenchymal Stromal Cells

ASC was isolated from fat tissue, which was kindly provided by the Clinic for Plastic, Hand and Burns Surgery (RWTH Aachen University Hospital, Aachen, Germany), after informed consent of the patient. Isolation was approved by the local ethics committee at the medical faculty of the RWTH Aachen University (EK 067/18). For digestion of fatty tissue, minced fat was mixed with Collagenase I (Sigma-Aldrich, Taufkirchen, Germany) and placed into gentleMACS Octo Dissociator (Miltenyi Biotec, Bergisch Gladbach, Germany) for 13 min and was then incubated for 30–40 min at 37 °C with gentle agitation on a roller mixer. To stop digestion, Dulbecco’s Modified Eagle Medium with 10% FCS and 1% ABM was added. Suspension was filtered through a 200 µm non-woven and centrifuged for 10 min at 300× *g*. After supernatant was removed, pellets were washed with PBS and centrifuged again. Pellet was resuspended, transferred into cell culture flasks, and cultured using Mesenpan medium (PAN-Biotech) with Mesenpan Growth Supplement (PAN-Biotech), 1% ABM and 2% FCS, incubated at 37 °C and 5% CO_2_. The first change of culture medium was done after 24 h, and afterward, it was changed every two days. At 70–80% confluency, cells were washed with PBS and passaged using Trypsin/EDTA. Experiments were performed using ASC in p2–5.

### 2.3. Cell Labeling

SK-MEL-28 and ASC were cultured in cell culture flasks (T25, Greiner Bio-One, Frickenhausen, Germany) to 80% confluence. Gd_2_O_3_-dex-RB nanoparticles were prepared and dispersed in modified DMEM or modified Mesenpan to label SK-MEL-28 or ASC, respectively. Cells were washed with PBS and Gd_2_O_3_-dex-RB nanoparticles in a 5 mL culture medium were pipetted into the cell culture flasks. Cells were incubated for 24 h at 37 °C and 5% CO_2_. To remove excessive nanoparticles, the cell culture medium was removed, and cells were washed three times with PBS.

SK-MEL-28 was incubated with Gd_2_O_3_-dex-RB nanoparticle concentrations varying from 0 mM up to 10 mM, and ASC from 0 mM up to 4 mM. Assessments were carried out with labeling concentrations as listed in Table 1.

### 2.4. Characterization

#### 2.4.1. Transmission Electron Microscopy

Gd_2_O_3_ nanoparticles, Gd_2_O_3_-dex nanoparticles, and Gd_2_O_3_-dex-RB nanoparticles samples were prepared in UPW (Table 2). Nanoparticles in solution were allowed to adsorb on glow discharged formvar–carbon-coated nickel grids (Maxtaform, 200 mesh, Plano, Wetzlar, Germany) for 10 min. Adhesive drops were removed with filter paper. Samples were viewed at an acceleration voltage of 100 kV using a Hitachi HT7800 (Hitachi, Düsseldorf, Germany) transmission electron microscope (TEM).

SK-MEL-28 and ASC were labeled (Table 1) and fixed in 3% glutaraldehyde in 0.1 M Soerensen’s phosphate buffer, scratched off from the tissue plate, and embedded in 5% low-melting agarose (Sigma-Aldrich). After post-fixation in 1% OsO_4_ (Carl Roth, Karlsruhe, Germany) in 25 mM sucrose buffer, samples were dehydrated by ascending ethanol series, incubated in propylene oxide (Serva, Heidelberg, Germany), and embedded in Epon (Serva). Ultrathin sections (70–100 nm) were cut and stained with 0.5% uranyl acetate and 1% lead citrate (both EMS, Hatfield, PA, USA) to enhance contrast. Samples were viewed at an acceleration voltage of 60 kV using a Zeiss Leo 906 (Carl Zeiss, Oberkochen, Germany) TEM.

#### 2.4.2. Zeta Potential

The surface charges of prepared nanoparticles (Table 2) were compared by measuring the respective surface zeta potentials (ZetaSizer Nano, Malvern Panalytical, Malvern, UK). Nanoparticles were dispersed in UPW as described (Table 2) and transferred into folded capillary zeta cells (Malvern, DTS1070). Measurements were performed at RT in triplicates for 12 runs each.

#### 2.4.3. Photometric Assessments

Fluorescence properties were assessed using a fluorescence microplate reader (Tecan Infinite M200, Tecan, Switzerland). For all measurements, a 96-well all-black microtiter plate (MTP, Costar, Thermo Fisher Scientific) was prepared with 200 µL of the examined sample including five technical replicates.

##### Coating Steps

MTPs with UPW, Gd_2_O_3_ nanoparticle, Gd_2_O_3_-dex nanoparticle, and Gd_2_O_3_-dex-RB nanoparticle samples (Table 2) were prepared. Fluorescence signal was measured after excitation at λ_ex_ = 538 nm at an emission wavelength λ_em_ = 576 nm.

##### Concentration Series

A dilution series of Gd_2_O_3_-dex-RB nanoparticles was prepared (Table 2). SK-MEL-28 and ASC were labeled (Table 1), trypsinized, and resuspended in modified DMEM (SK-MEL-28) or modified Mesenpan (ASC). Fluorescence signal was measured after excitation at λ_ex_ = 538 nm at an emission wavelength λ_em_ = 576 nm.

##### Spectrum

Microtiter plates with 0.1% rose bengal in UPW and Gd_2_O_3_-dex-RB nanoparticle samples (Table 2) were prepared. SK-MEL-28 and ASC were labeled (Table 1), trypsinized, and resuspended in modified DMEM (SK-MEL-28) or modified Mesenpan (ASC). The excitation spectrum was identified after excitation between λ_ex_ = 400 nm and λ_ex_ = 560 nm at an emission wavelength λ_em_ = 576 nm. The emission spectrum was identified between λ_em_ = 560 nm and λ_em_ = 700 nm after excitation at λ_ex_ = 538 nm.

#### 2.4.4. Fluorescence Imaging

To assess the image properties of Gd_2_O_3_ nanoparticles, Gd_2_O_3_-dex nanoparticles, and Gd_2_O_3_-dex-RB nanoparticles, samples (Table 2) of 100 µL each were pipetted on a microscope slide and covered with a cover glass. A fluorescence microscope (Zeiss Observer.Z1, Carl Zeiss) with a camera (AxioCam MR3, Carl Zeiss) was used for imaging the samples in bright-field and fluorescence microscopy after excitation at λ_ex_ = 546/12 nm at an emission wavelength λ_em_ = 575–640 nm. Images were taken with a magnification of 10× and an exposure time of 150 ms.

SK-MEL-28 and ASC were labeled (Table 1) and directly imaged in the flasks by bright-field and fluorescence microscopy at an exposure time of 300 ms.

#### 2.4.5. MRI Measurement

MRI measurements were performed using a clinical MRI Scanner (3 T Achieva, Philips Healthcare, Amsterdam, The Netherlands). All samples were embedded inside a phantom containing polyacrylic acid and sodium chloride according to ASTM standards [14]. Each Gd_2_O_3_-dex-RB nanoparticle sample (Table 2) was prepared for 6 different Gd_2_O_3_ concentrations *c* up to 10 mMol. For investigation of the longitudinal and transverse relaxation rates (*R*_1_ and *R*_2_), an Inversion Recovery Spin Echo (IRSE) sequence (repetition time TR = 5000 ms, 1 echo, echo time TE = 12 ms, inversion delay times TI = [100, 400, 700, 1000, 1300, 1600, 1900, 2200, 2500] ms, field of view FOV = 320 × 320 mm, matrix 256 × 256, slice thickness 5 mm, flip angle 90°) and a Turbo Spin Echo (TSE) sequence (TR = 1500 ms, 32 echoes, TE = [10, 20, …, 320] ms, FOV = 366 × 366 mm, matrix 512 × 512, slice thickness 3 mm, flip angle 90°) were used, respectively. From these measurements, the longitudinal and transversal relaxivities, *r*_1_ = (*R*_1_ − *R*_1,0_)/*c* and *r*_2_ = (*R*_2_ − *R*_2,0_)/*c*, of the samples were calculated using MATLAB (The MathWorks, Natick, MA, USA). *R*_1,0_ and *R*_2,0_ denote the relaxation rates of the DMEM medium which were used as reference. For cell experiments, SK-MEL-28 was labeled with Gd_2_O_3_-dex-RB nanoparticles (1 mMol) (Table 1), trypsinized, and resuspended in modified DMEM. Samples of labeled cells were prepared in a concentration series from 500,000 cells up to 4 × 10^6^ cells per mL and measured with TSE and IRSE sequences. A sample of 4 × 10^6^ unlabeled cells per mL served as a control.

#### 2.4.6. Cell Proliferation

SK-MEL-28 and ASC were labeled (Table 1), trypsinized, and seeded with 5700 cells/cm^2^ into 12-well plates (VWR International, Langenfeld, Germany). Proliferation was assessed on SK-MEL-28 using three technical replicates of each labeling concentration, on ASC using one well per donor (*n* = 3 independent donors). All well plates were incubated at 37 °C and 5% CO_2_. Culture medium was changed daily. Cells were examined on days 1, 3, and 7. On each examination day, the cells to be assessed were washed with PBS and fixed using ice-cold 100% methanol (VWR International). Fixed cells were stored at 4 °C until analysis. Cell proliferation was quantified by 4′-6′-diamidino-2-phenylindole (DAPI) staining. The cells were stained using 1 mg/mL DAPI in PBS. DAPI fluorescence was analyzed by excitation at 365 nm and emission at 445/50 nm with a fluorescence microscope. Five images were taken following a specific layout with a magnification of 4×. Counting of DAPI-stained cell nuclei was done using the software CellProfiler3.1.9 [15].

#### 2.4.7. Statistical Evaluation

All photometric data were processed with Microsoft Excel 2016. Diagrams, curve fitting, and statistical evaluation were accomplished using GraphPad Prism Version 9. For statistical evaluation of fluorescence signal after nanoparticle coating steps, one way-ANOVA was used with Tukey’s post hoc test for multiple comparisons. A *p*-value below 0.05 was considered significant. For curve fitting, a non-linear fit with hyperbola shape was selected.

For cell number evaluation, cell counts were converted into cells/mm^2^. The significance of the 0 mM control was tested with ordinary one-way ANOVA with Dunnett’s multiple comparisons test. To compare the proliferation, cell counts were normalized to the mean value of the respective concentration of day 1. The significance of the 0 mM control was tested with two-way ANOVA with Šídák’s post hoc test for multiple comparisons. A *p*-value below 0.05 was considered significant.

## 3. Results

### 3.1. Gd_2_O_3_-dex-RB Nanoparticles

#### 3.1.1. TEM

To characterize shape and size, the nanoparticles were observed by TEM (Gd_2_O_3_ nanoparticles, Gd_2_O_3_-dex nanoparticles, and Gd_2_O_3_-dex-RB nanoparticles, Figure 2). Along the process of sample preparation and drying, the particles agglomerated, which made the measurement of a single particle more difficult. Nevertheless, it was shown that at all coating steps, the particles appeared to be oblong and were approximately 23 nm to 35 nm in diameter and up to 60 nm in length, which is close to the manufacturer’s labeling (20–40 nm) [16]. Additionally, the size did not change independent of the coating steps, which indicates TEM is neither able to detect dextran nor rose bengal.

#### 3.1.2. Zeta Potential

Zeta potentials were measured for Gd_2_O_3_, Gd_2_O_3_-dex, and Gd_2_O_3_-dex-RB nanoparticles. Native Gd_2_O_3_ nanoparticles’ zeta potential was 19.8 mV (Appendix A). Upon coating with dextran (Gd_2_O_3_-dex nanoparticles), the zeta potential shifted to 11.1 mV (Appendix A). A decrease in zeta potential can be explained by the increased presence of hydroxy groups within the polymeric coating.

Zeta potential decreased further to −3.52 mV for Gd_2_O_3_-dex-RB nanoparticles (Appendix A). Rose bengal is an anionic dye, hence, decreasing the zeta surface charge of the particles. Furthermore, an increase in surface-exposed hydroxy groups in rose bengal compared to dextran explains the observed decrease in zeta surface charge.

#### 3.1.3. Photometric Assessments

To evaluate the fluorescence properties of uncoated and coated nanoparticles, photometric measurements were accomplished. The emission and excitation spectra of Gd_2_O_3_-dex-RB nanoparticles compared to rose bengal can be seen in Figure 3a,b. Rose bengal showed the highest fluorescence signal at an emission of 576 nm and an excitation of 538 nm. Gd_2_O_3_-dex-RB nanoparticles showed the highest fluorescence signal at an emission of 570 nm and an excitation of 550 nm. Comparing the emission and excitation spectra of coated nanoparticles with the spectra of rose bengal, a similar trend can be observed. This proves an effective fluorescence coating and the presence of rose bengal as the working agent. Further photometric assessments were done using an excitation of λ_ex_ = 538 nm and an emission of λ_em_ = 576 nm.

The fluorescence signal of the bare nanoparticles after the two coating steps is displayed in Figure 3c. Both bare Gd_2_O_3_ nanoparticles and dextran coating did not show any significant emissions. After adding rose bengal, the nanoparticles showed a significant fluorescence signal.

Measuring a concentration series of Gd_2_O_3_-dex-RB nanoparticles, a hyperbolic slope could be observed (Figure 3d). The fluorescence signal increased with increasing nanoparticle concentration. Following hyperbolic increases, it is expected to reach a plateau. Here, the saturation concentration is not yet reached with the concentrations used.

#### 3.1.4. Fluorescence Imaging

The photometric results were confirmed by fluorescence microscopy. In Figure 4, the imaging properties of Gd_2_O_3_ nanoparticles, Gd_2_O_3_-dex nanoparticles, and Gd_2_O_3_-dex-RB nanoparticles are demonstrated. Nanoparticles of all coating steps can be detected using bright-field microscopy; they accumulate at all stages. Fluorescence signaling was only detected for Gd_2_O_3_-dex-RB nanoparticles. The fluorescence signal overlaps with nanoparticles detected in the bright-field channel. The more nanoparticles accumulated, the higher the fluorescence signal.

#### 3.1.5. MRI Measurement

Figure 5 shows that longitudinal and transversal relaxation rates of Gd_2_O_3_-dex-RB nanoparticles linearly increase with Gd_2_O_3_ concentration. Exemplary MR images of Gd_2_O_3_-dex-RB nanoparticle samples measured with IRSE and TSE sequences show, respectively, a brighter and darker contrast with increasing concentration. These results demonstrate that Gd_2_O_3_-dex-RB nanoparticles can, in principle, be used as both positive and negative contrast agents. The values of longitudinal and transversal relaxivities are (0.08 ± 0.02) mMol^−1^s^−1^ and (3.03 ± 0.86) mMol^−1^s^−1^, respectively.

### 3.2. Cell Labeling

As the next step, cells were incubated with Gd_2_O_3_-dex-RB nanoparticles for further examination of labeling properties and cell viability. SK-MEL-28 were incubated with 0, 0.5, 1, 2, 4, 8, and 10 mM. ASC were incubated with 0, 1, 2, and 4 mM. Incubation time was 24 h. Examinations included photometric assessments, fluorescence microscopy, TEM, and MR imaging as well as cell viability assessments observing cell proliferation for 7 days after incubation.

#### 3.2.1. Photometric Assessments

The fluorescence signals of SK-MEL-28 and ASC, which were incubated with ascending concentrations of Gd_2_O_3_-dex-RB nanoparticles, are shown in Figure 3e,f. Cells without labeling did not show any signal. The fluorescence signal of the labeled cells increased with increasing nanoparticle concentrations, which supports the labeling process being effective. Because the cells gained fluorescence signals as the nanoparticle concentration increased, there is evidence of a growing uptake rate in the cells. Nevertheless, the gain of signal did not increase proportionally to the nanoparticle concentration. The slope ran hyperbolic, which indicates that a saturation concentration can be reached for the cell’s intake. The concentrations of Gd_2_O_3_-dex-RB nanoparticles used to label SK-MEL-28 seemed to be closer to the plateau because the increase in fluorescence signal already reduced noticeably.

The emission and excitation spectra of labeled SK-MEL-28 and ASC are shown in Figure 3a,b. The peaks of emission for both cell lines are shifted to 592 nm compared to rose bengal’s peak of emission located at 576 nm. Descending from the peaks, the curves converge to nearly the same regression. The progression of the excitation curve of labeled SK-MEL-28 shows to be lower than rose bengal but then converges to the latter at 500 nm. Labeled ASC shows a lower excitation curve than rose bengal although the progression of the curve appears to be very similar. For both SK-MEL-28 and ASC, the peak of excitation is reached at an emission of 538 nm, which is the same as rose bengal. It proves that Gd_2_O_3_-dex-RB nanoparticles were present in the sample and rose bengal is working effectively as the fluorescent agent.

#### 3.2.2. Fluorescence Imaging

Representative images of SK-MEL-28 and ASC incubated in ascending concentrations of Gd_2_O_3_-dex-RB nanoparticles are shown in Figure 6 and Figure 7. Cells that were not incubated with nanoparticles were detected in bright-field but not in fluorescence microscopy. For cells that were incubated with either nanoparticle concentration, a fluorescence signal was observed. With increasing nanoparticle concentrations, cells emitted brighter fluorescence signals. This suggests that the amount of internalized nanoparticles can be increased by incubating the cells with higher nanoparticle concentrations.

ASC were smaller but labeling signals are as bright as the ones of SK-MEL-28. As SK-MEL-28 labeled with 0.5 mM nanoparticles did show only a low fluorescence signal, this concentration was not chosen to be tested with ASC. In addition, the concentration of 8 mM nanoparticles was also not included in the assessment of ASC. A bright fluorescence signal was already achieved using 4 mM and ASC seemed to be more sensitive to increasing concentrations.

In higher magnification, only the cells’ cytosol was found to be fluorescing, and the nucleus was spared such that the cell bodies and cell extensions were clearly distinguishable from the nucleus and the surroundings. Representative images of a close-up view can be seen in Appendix A. It suggests that the nanoparticles are only internalized into the cytosol and are not able to enter the nucleus of the cells.

In addition to the internalized nanoparticles, there were also nanoparticles found outside and around the cells, which could be observed especially by looking at images of higher concentrations.

#### 3.2.3. Transmission Electron Microscopy

Because nanoparticles were found inside as well as around and outside the cells, TEM was chosen to examine the exact location of the nanoparticles. TEM imaging of labeled SK-MEL-28 (Figure 8) and ASC (Figure 9) brought evidence that nanoparticles were internalized into the cells. The nanoparticles are accumulated in the cytosol in circular shapes (ii) having a thin membrane (iii) around them. This suggests that the nanoparticles are located inside vesicles. There are several vesicles filled with nanoparticles distributed in the cytosol. Nanoparticles outside of the cells were also observed (i). They appeared to be close to the cell’s membrane.

#### 3.2.4. MRI Measurement

Figure 10 shows exemplary MRI images of SK-MEL-28 cells labeled with 1 mM Gd_2_O_3_-dex-RB nanoparticles. Images measured with a TSE sequence (Figure 10a) show an increasingly dark contrast for higher amounts of up to 4 × 10^6^ cells per mL. In the images measured with an IRSE sequence, no contrast difference with respect to the control could be observed for increasing cell numbers (Figure 10b).

#### 3.2.5. Cell Proliferation

To examine the impact of Gd_2_O_3_-dex-RB nanoparticles on the cell proliferation rate, SK-MEL-28 was labeled, and cell count was assessed on day 1, day 3, and day 7 after incubation. SK-MEL-28 was incubated with 0, 0.5, 1, 2, 4, and 8 mM nanoparticles. The cell count of the different concentrations on day 1 can be seen in Figure 11a. Cells labeled with 0.5, 1, 4, and 8 mM show a significant reduction compared to unlabeled cells. The lowest cell count was at 8 mM. This suggests that Gd_2_O_3_-dex-RB nanoparticles have an impact on existing cells directly after incubation was processed. The higher the concentration of nanoparticles, the bigger the impact gets. Moreover, 8 mM seems to cross a concentration threshold, which causes a significant reduction of cell number to nearly zero cells.

A comparison of cell proliferation normalized to the mean value of the respective concentration of day 1 can be found in Figure 11c. Comparing all concentrations on day 3, cell proliferation is decreasing with increasing nanoparticle concentration, showing a significant reduction for high concentrations of 4 and 8 mM. Labeling concentrations of 0 mM and 1 mM reach up to 500% on day 7. 0.5, 2, and 4 mM show a proliferation of 300 to 350%. Cells incubated with 8 mM seem to stop proliferation because the cell count decreases from day 1 until day 7. All concentrations except 8 mM show an increase in cell number from day 1 to day 7. This suggests that even though the nanoparticles have an impact directly after incubation, the surviving SK-MEL-28 still has the potential to proliferate. The same as described before, 8 mM seems to have crossed the threshold because the cells do not proliferate, and cell number reduces.

ASC was examined the same way after incubation with 0, 1, 2, and 4 mM Gd_2_O_3_-dex-RB nanoparticles. The cell count on day 1 can be seen in Figure 11b. Labeling concentrations of 1 and 2 mM show a slight reduction of cell count; 4 mM samples show a significant reduction. Gd_2_O_3_-dex-RB nanoparticles seem to have an impact on these cells crossing a threshold of 4 mM.

Looking at Figure 11d, the proliferation is shown for all labeling concentrations on day 3 and day 7 normalized to the mean value of the respective concentration on day 1. Cells incubated with 1 mM show a high increase in cell number up to approximately 1100%, which is comparable to the unlabeled control reaching 900%. Cells labeled with 2 mM show an increasing cell number from day 1 to day 7, but the increase is significantly lower than the unlabeled cells. Cells labeled with 4 mM do not show an increase in cell number. The reduction is significant and cell number is reduced to almost 0% at day 7. Therefore, 4 mM seems to cross the concentration threshold, at which ASC are not able to proliferate anymore.

## 4. Discussion

For a more efficient synthesis of Gd_2_O_3_-dex-RB nanoparticles, a new protocol was created, which shortens the incubation times of coating steps from 24 h down to 1 h per coating [10]. This allows for achieving full nanoparticle modification in less than one day. In this study, the new protocol for Gd_2_O_3_-dex-RB nanoparticles was tested for a successful synthesis of nanoparticles, which shows a good performance in fluorescence microscopy and MRI and can efficiently be used to label cells showing low toxic effects.

Native Gd_2_O_3_ nanoparticles’ zeta potential was 19.8 mV, which is in accordance with previously reported zeta potentials for nanoparticles consisting of Gd_2_O_3_ [17]. The zeta potential of Gd_2_O_3_-dex nanoparticles shifted to 11.1 mV. The decrease in zeta potential upon the addition of dextran is also in accordance with prior reports from Gardner [18]. Zeta potential decreased further to −3.52 mV for Gd_2_O_3_-dex-RB nanoparticles. Purchased Gd_2_O_3_ nanoparticles exhibited zeta potentials in agreement with published values, proofing effective nanoparticle formation and surface charge. Dextran coating was shown indirectly by a decrease in zeta potential compared to native Gd_2_O_3_ nanoparticles. Gd_2_O_3_-dex nanoparticles still exhibit a positive surface charge. The anionic nature of the rose bengal dye primarily allows it to adhere to the positively surface charged Gd_2_O_3_-dex nanoparticle and further causes a decrease in nanoparticle surface potential. Finally, zeta potential measurements also indicate the presence of rose bengal dye on the surface of nanoparticles after addition.

For detailed information about the phase composition of Gd_2_O_3_-dex-RB nanoparticles and further confirmation of successful functionalization by using dextran and rose bengal as coating, measurements of the electron diffraction would be useful in future studies.

Gd_2_O_3_-dex-RB nanoparticles produced with the new protocol showed fluorescence properties, as expected. Rose bengal works as the fluorescent agent, the signal increases with higher concentrations of nanoparticles, and the excitation and emission spectra are similar to the ones of rose bengal alone. SK-MEL-28 and ASC were able to internalize Gd_2_O_3_-dex-RB nanoparticles and showed fluorescence signals, as expected, afterward. Excitation and emission spectra were slightly moved but still fell into the same range as rose bengal.

The more Gd_2_O_3_-dex-RB nanoparticles were present in the cell medium, the more nanoparticles were taken up by the cells. However, the nanoparticle uptake rate did not run proportional to the exposure concentration. The curve ran hyperbolic, indicating a maximum of nanoparticles that is possible to be internalized. Therefore, it is suggested that the cells’ uptake rate in 24 h is limited. Klasson et al. studied the labeling of THP-1 cells with Gd_2_O_3_-DEG nanoparticles [19]. The number of nanoparticles, which were internalized, increased with increasing nanoparticle concentration for low concentrations. Samples incubated with 0.5 mM showed an uptake of 8% of the total exposure. Samples incubated with 2.5 mM showed an uptake of 37%. Moreover, they also found that the uptake rate did not run proportional to the total concentration in the cell medium. The higher the exposure concentration, the more the slope is reduced, which agrees with our observation of a limited uptake rate. These observations can be explained by either an equilibrium between endocytosis and exocytosis [20] or by the limited capacity of particle uptake into the cells [21]. In addition to that, the time scale of endocytosis and exocytosis is also influenced by several parameters, such as digestion or translocation within cells [20]. Slabu et al. established a nanoparticle uptake fitting model which allows us to determine the endocytosis and exocytosis rates. Models such as these could be applied in future studies for determining the uptake rates of Gd_2_O_3_-dex-RB nanoparticles for different incubation times and different cell lines [22].

In fluorescence microscopy, Gd_2_O_3_-dex-RB nanoparticle-labeled SK-MEL-28 and ASC showed a noticeable fluorescence signal. All cells in one sample seemed to take in similar amounts of nanoparticles because they showed similar brightness in the imaging process. Comparing both cell types, ASC seemed to take in more nanoparticles in 24 h because lower concentrations already showed bright fluorescence signals in fluorescence microscopy for the same imaging conditions. In addition, it was shown that the nanoparticles are internalized into the cytosol of the cells. The area of the nucleus does not show any fluorescence signal. Thus, the pathway of internalization seems to only work for crossing the cell’s membrane.

Furthermore, fluorescence microscopy as well as TEM results showed that there are nanoparticles present outside the cells. This can be observed especially at higher concentrations. Most nanoparticles were located near the cell membranes. The nanoparticles settled and seemed to stick to the cell membranes. As the cells need to bind the nanoparticle to their surface before internalizing them [23], this seems like a step that is not avoided along the labeling process. Therefore, it is possible that some of the measured Gd_2_O_3_-dex-RB nanoparticles were not in the cytosol but outside attached to the cells. Similar findings were described by Hedlund et al. [24].

However, in this study, it was proven that Gd_2_O_3_-dex-RB nanoparticles were inside the cells. In TEM imaging, nanoparticles were found inside vesicles in the cytosol. The intracellular vesicles suggest an endocytosis-based uptake pathway [25]. This agrees with several studies showing endocytosis as the uptake pathway of similar nanoparticles [24,26,27]. As described by Donahue et al., there are several different endocytosis-based uptake pathways [25]. As neither SK-MEL-28 nor ASC is naturally phagocytic, phagocytosis is unlikely to be the uptake pathway. Shi et al. labeled human bone marrow-derived mesenchymal stem/stromal cells (MSC) with bifunctional Eu^3+^-doped Gd_2_O_3_ nanoparticles [27]. They observed a non-specific endocytotic pathway for nanoparticle uptake. Santelli et al. labeled MSC with Gd_2_O_2_S:Eu^3+^ nanoparticles [26]. They suggest micropinocytosis as a pathway of nanoparticle internalization. Moreover, the interactions of Gd_2_O_3_-dex-RB nanoparticles with the cell culture medium probably lead to the formation of a protein corona, as well as affect their uptake behavior and pathway [28]. Thus, the authors conclude that different uptake pathways might be influenced by the biological and natural differences of the cells as well as the different labeling methods. The exact uptake pathway for Gd_2_O_3_-dex-RB nanoparticles was not further examined, as it was not the focus of this study. In future studies, this could be focused on dynamic light scattering (DLS) measurements to collect data on the nanoparticles’ hydrodynamic sizes, confocal laser scanning microscopy, TEM, and/or refractive index-based spectroscopic methods [29]. However, especially for DLS, the non-spherical shape can cause problems. Like all particle-sizing techniques, DLS is not able to adequately describe an oblong particle such as the Gd_2_O_3_ nanoparticles used in this study. Their result would be the same as a particle that has the same translational diffusion speed but cannot be considered correct. In addition, nanoparticle agglomeration can add another difficulty to produce reliable data using DLS.

Indeed, during all coating steps, agglomeration of nanoparticles could be observed. The tendency of nanoparticles to agglomerate in dry form as well as in suspension can be explained by van der Waal’s forces [30]. Because the formation of clots affects further coating steps, the use of an ultrasound bath was introduced between all coating steps and before applying the nanoparticles onto the cells. Nevertheless, the formation of clots could especially be observed in cell culture flasks. Nanoparticles are described to change their dispersity behavior as they interact with the surrounding cell culture medium [28]. Proteins are adsorbed onto the surface of the nanoparticles because of different interactions due to electrostatic, hydrogen, and hydrophobic properties. This causes a change in the surface charge of the nanoparticles. The more it moves towards the value zero by protein adsorption, the repulsive forces between nanoparticles are reduced. Gravitational forces lead to the settling of the nanoparticles and van der Waal’s forces can act stronger, leading to more agglomeration [30]. Nevertheless, the internalization of the nanoparticles does not seem to be disturbed by the agglomeration. Possibly, internalization by the cells starts as soon as the nanoparticles are present, and the agglomeration is not as pronounced as after 24 h. In addition, even agglomerated nanoparticles could be internalized, because TEM showed several nanoparticles being present in one vesicle. Moreover, in a study by Santelli et al., nanoparticles measuring up to 465 nm were found to be taken up by cells and sizes larger than 200 nm even induced an uptake of higher concentrations of nanoparticles into the cells [26]. This suggests that the agglomeration of the nanoparticle does not interfere with internalization.

Experiments collecting data about functionalized nanoparticles were always carried out directly after synthesis was completed. In order to introduce Gd_2_O_3_-dex-RB nanoparticles as a common method for dual cell labeling, further studies must be carried out to examine the shelf life of the suspensions produced. Collecting more information about their short-term and middle-term stability is essential to maintain ease of use.

The ratio of the relaxivities *r*_2_/*r*_1_ of the Gd_2_O_3_-dex-RB nanoparticles is approximately 38. This relatively high value indicates a stronger shortening effect of T2 relaxation and thus a better suitability for the use of Gd_2_O_3_-dex-RB nanoparticles as a negative MRI contrast agent. In comparison, smaller ratios of approximately *r*_2_/*r*_1_ ≤ 7 were reported for gadolinium nanoparticles, which were investigated for their use as positive MRI contrast agents [24,27,31,32,33,34]. Normally, paramagnetic Gd^3+^ ions on the surface of the particles are responsible for the ability to generate positive contrast in MRI. The large magnetic moments of the Gd^3+^ ions shorten the longitudinal relaxation of the water protons in the vicinity of the particles [31,35]. This effect is more pronounced for ultra-small particles (size approx. 1–5 nm) due to their high surface-to-volume ratio. Large or agglomerated particles, e.g., by encapsulation inside lysosomes during internalization in cells, show lower surface-to-volume ratios resulting in lower *r*_1_ values and, hence, in higher *r*_2_/*r*_1_ ratios [24,27,36]. In the same way, the Gd_2_O_3_ particle agglomerates observed in this study (cf. Figure 2) could be responsible for their relatively high *r*_2_/*r*_1_ ratio. Furthermore, a critical Gd concentration for optimal *T*_1_ enhancement has been reported in several studies in the literature [19,37,38]. For concentrations above this threshold (in the range of 0.1–0.6 mMol), *T*_2_ relaxation times become dominant leading to a signal loss, which is characteristic of negative contrast agents. For this reason, the *T*_1_ enhancement for the Gd_2_O_3_-dex-RB particles in this study is probably best at lower concentrations. This behavior is consistent with the dark contrast of the images of labeled SK-MEL-28 cells measured with a TSE sequence (cf. Figure 10).

Cell proliferation assessments showed that SK-MEL-28 and ASC are not influenced by the Gd_2_O_3_-dex-RB nanoparticles in lower concentrations. At a concentration of 8 mM, SK-MEL-28 showed a significant reduction of cell number and a proliferation stop with further loss of cells. The concentration threshold of ASC seems to be lower as they show the reduced cell number and a stop of proliferation at 4 mM already.

The authors assume different reasons for the impact of Gd_2_O_3_-dex-RB nanoparticles on the cells. After incubation, the nanoparticles sediment onto the cells and the bottom of the cell culture flasks. Higher nanoparticle concentrations might cover the cells and limit their ability to communicate with their environment as well as their nutrient intake. Thus, this can lead to a lack of cell–cell interactions and a level of nutrition concentration which is too low to maintain normal cell activity. If this situation is persisting for too long, it leads to inhibition of proliferation and cell death [39]. In addition, gadolinium and rose bengal are both toxic in higher concentrations [40,41] such that the intake of too high concentrations of Gd_2_O_3_-dex-RB nanoparticles could lead to cell death caused by toxicity. Moreover, close attention needs to be paid to the risk of the release of highly toxic free Gd^3+^ ions. Studies showed that free Gd^3+^ ions interact with different kinds of body tissues. Interactions with renal tissues can lead to systemic nephrogenic fibrosis (NFS) after repeated use [42]. According to the manufacturer’s labeling, Gd_2_O_3_ nanoparticles are soluble in acid but insoluble in water [16]. There was no more precise data given about a certain pH range. Functionalization and experiments for characterization were always carried out in UPW with pH 5.95 and cell experiments were always carried out in a cell culture medium with a physiological pH. As the proliferation of SK-MEL-28 and ASC are not influenced by the Gd_2_O_3_-dex-RB nanoparticles in concentrations below 8 and 4 mM, respectively, the formation of free Gd^3+^ ions does not present a relevant problem while using these concentrations. Nevertheless, an assessment of existing free Gd^3+^ ions is needed before establishing in vivo experiments.

Several studies using Gd_2_O_3_ nanoparticles with varying modifications assessed cell viability and proliferation after incubation. Shi et al. proved non-toxicity for 0.5 mM Eu3+-doped Gd_2_O_3_ nanoparticles up to seven days after incubation of MSC [27]. Klasson et al. found that Gd_2_O_3_-DEG nanoparticles in concentrations up to 2.5 mM are non-toxic for THP-1 cells [19]. Fang et al. studied the viability of HK-2 proximal tubule epithelial cells 24 and 48 h after incubating the cells with Gd_2_O_3_ and Gd_2_O_3_-PVP nanoparticles [43]. The tested concentrations from 0.16 mM to 1.6 mM were found to be non-toxic for the cells. All studies mentioned above did only evaluate cell behavior after labeling with the indicated concentrations which they found to be non-toxic. Higher concentrations that might have a toxic effect on the cells were not examined or shown. Bennewitz et al. tested poly(lactic-co-glycolic acid) (PLGA) encapsulated Gd_2_O_3_ nanoparticles on mouse embryonic fibroblast cells and evaluated the viability after 48 h [8]. The amount of nanoparticles they incubated the cells with ranged from 12.5 µg/mL to 1.6 mg/mL. A loss of viability occurred for concentrations higher than 200 µg/mL. As a different unit is used to indicate labeling concentrations, nanoparticles are modified differently and cells varied as well, and it is hard to directly compare their findings to the results reported in this current study. Nevertheless, the loss of viability at a certain concentration threshold is in accordance with our findings.

Varying labeling methods, cell lines, and exposure times of the nanoparticles to the cells in all these studies make it difficult to compare. Most studies only reported incubation concentrations that are lower than the toxicity threshold. However, studying the limitations of different cell lines and nanoparticles is important for future applications.

In this study, an approach was chosen in which Gd_2_O_3_ as an MRI contrast agent was used as the basis material for further functionalization. The functionalization aimed to add fluorescence properties (RB) and improved biocompatibility (dex). Recently, several studies have been published in which Gd_2_O_3_ was used for cell labeling and tracing via MRI [17,26,44,45]. However, there are also various different approaches containing gadolinium as an MRI contrast agent but focusing on other basis materials, e.g., Popov et al. used cerium oxide nanoparticles which were doped with gadolinium [46]. The authors decided on working with cerium oxide as it was shown to be fully compatible with cells and tissue and added Gd^3+^ as an MRI contrast agent. No further functionalization regarding fluorescence properties was carried out. High relaxivity values as well as low cell toxicity for human mesenchymal stem cells were described. Another approach involved the preparation of carbon dots which Zheng et al. doped with gadolinium [47]. Carbon dots were used in order to achieve lower biological toxicity caused by the leakage of free Gd^3+^ ions. The authors describe high *r*_1_ relaxivity values as well as bright fluorescence properties. Even at high concentrations, Gd-doped carbon dots showed low long-term toxicity. Many other approaches to the use of gadolinium have been published and will be further explored in the future. It remains exciting to strive for ever-better biocompatibility and ever-better imaging properties.

In this study, it was shown that the use of Gd_2_O_3_-dex-RB nanoparticles is an adequate labeling method to track SK-MEL-28 and ASC in fluorescence microscopy and MRI. It can be useful in answering questions about cell locations and distributions in vitro and in vivo.

Fluorescence microscopy is a suitable tool for in vitro studies of cell locations. It is a standard method in most laboratories, such that the availability is good, and the equipment is relatively cheap compared to more complex methods. It provides an assessment with high sensitivity so that single cells can be detected in the sample. In addition, it is possible to be applied clinically for intraoperative imaging or endoscopy. The biggest disadvantage of fluorescence microscopy is the very small penetration depth. Because of light absorption and scattering, the emitted light in the visible spectrum reaches only a limited tissue penetration of several hundred micrometers. Therefore, it is not applicable for whole-body or even whole-organ scans of humans or bigger animals. Only small animals and surface structures are possible to assess with fluorescence microscopy imaging. As the histological detection of the cells has to happen ex vivo, this method is mostly not the appropriate choice for in vivo studies. In addition, spatial resolution is limited because only small areas can be assessed at a time. Another limitation of Gd_2_O_3_-dex-RB nanoparticles themselves is the possibility of photobleaching. The particles might not show enough fluorescence signal after some time. The assessment is therefore restricted to a certain time frame after applying the labeled cells to the sample.

MRI is a more applicable method for assessments of deeper compartments in the body and whole-body scans. The localization of cells in vivo is more feasible in MRI than in fluorescence microscopy. The spatial resolution is better because bigger areas can be examined, and even three-dimensional reconstruction is possible. Nevertheless, MRI has disadvantages, too. The equipment is big and expensive, such that the availability is limited. Moreover, the assessment was relatively insensitive. Single-labeled cells did not create a change in the MRI signal, which is large enough for their visualization. Higher numbers of cells were needed to deliver a sufficient signal change in MRI. Dealing with experiments in which single-cell detection is necessary, the usage of different contrast media such as nanometer-sized ultrasmall iron oxide particles (USPIOs) or micrometer-sized iron oxide particles (MPIOs) is applicable [48,49,50].

The combination of fluorescence microscopy and MRI produces complementary results to compensate for the disadvantages of each method and gives sufficient information about cell location and distribution.

## 5. Conclusions

In this study, a more efficient protocol for the production of Gd_2_O_3_-dex-RB nanoparticles was proven to be successful and allows an easy synthesis within only 4 h. The nanoparticles were demonstrated to be an adequate labeling agent to track SK-MEL-28 and primary cells as ASC in vitro via fluorescence microscopy and MRI without significant loss in cell viability. In the future, in vivo studies should be carried out for further proof of cell tracking properties.

## Figures and Tables

**Figure 1 nanomaterials-13-01869-f001:**
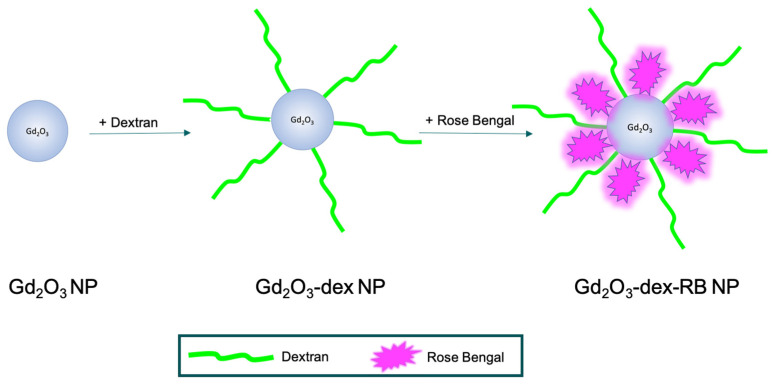
Illustration of Gadolinium(III)-oxide nanoparticle (Gd_2_O_3_ NP) functionalizing with dextran (dex) and rose bengal (RB).

**Figure 2 nanomaterials-13-01869-f002:**
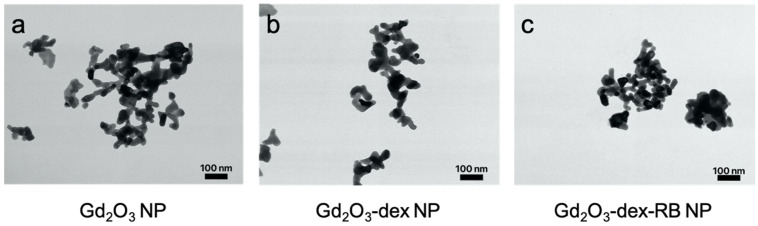
Transmission electron microscopy (TEM) imaging of Gd_2_O_3_ nanoparticles (**a**), Gd_2_O_3_-dex nanoparticles (**b**), and Gd_2_O_3_-dex-RB nanoparticles (**c**). Representative images, which show a typical diameter are shown.

**Figure 3 nanomaterials-13-01869-f003:**
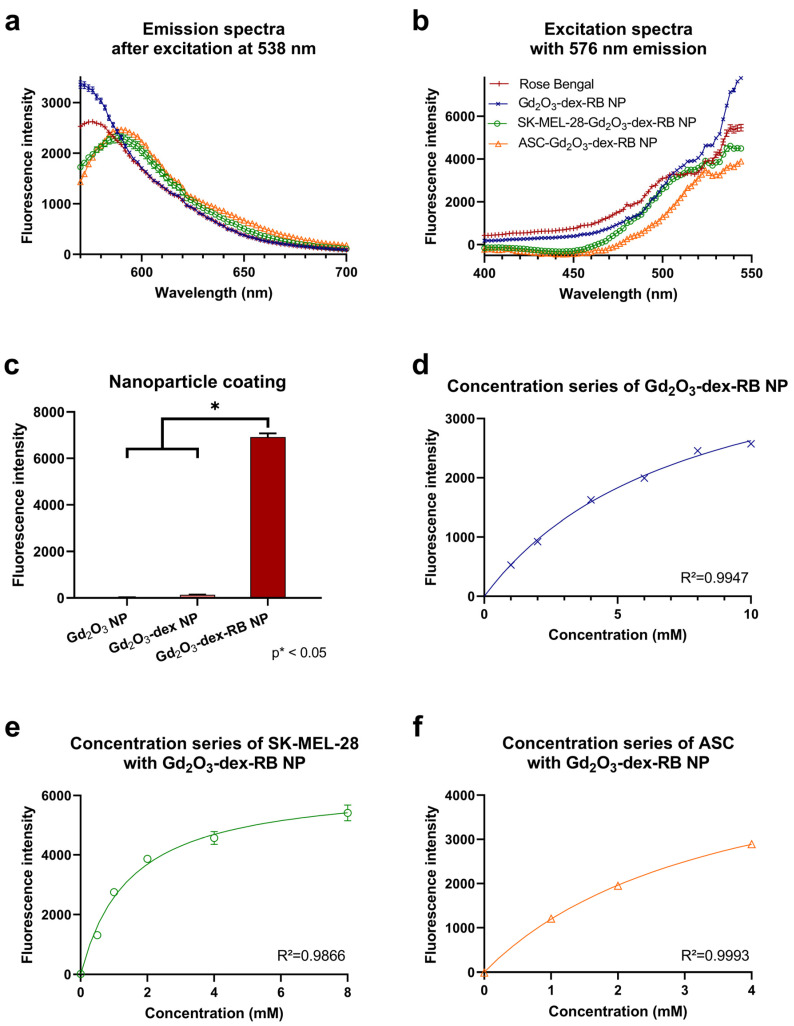
Emission spectra of Gd_2_O_3_-dex-RB nanoparticles and 0.1% rose bengal after excitation at 538 nm with *n* = 5 (**a**). Excitation spectra of Gd_2_O_3_-dex-RB nanoparticles and 0.1% rose bengal at 576 nm emission with *n* = 5 (**b**). Fluorescence signal of Gd_2_O_3_-dex-RB nanoparticles while proceeding through the coating steps (**c**) and of different concentrations of Gd_2_O_3_-dex-RB nanoparticles (**d**) with *n* = 5. Fluorescence signal of SK-MEL-28 (**e**) and ASC (**f**) which were labeled with increasing concentrations of Gd_2_O_3_-dex-RB nanoparticles with *n* = 5. Emission spectra of SK-MEL-28 and ASC labeled with Gd_2_O_3_-dex-RB nanoparticles in comparison to rose bengal with *n* = 5 (**a**). Excitation spectra of SK-MEL-28 and ASC labeled with Gd_2_O_3_-dex-RB nanoparticles in comparison to rose bengal with *n* = 5 (**b**).

**Figure 4 nanomaterials-13-01869-f004:**
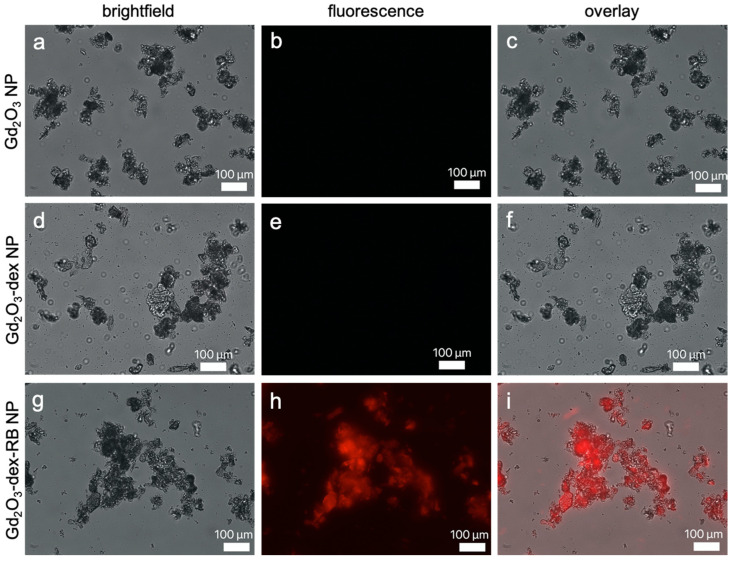
Microscopy of Gd_2_O_3_ nanoparticles (**a**–**c**), Gd_2_O_3_-dex nanoparticles (**d**–**f**), and Gd_2_O_3_-dex-RB nanoparticles (**g**–**i**) using bright-field (**a**,**d**,**g**), fluorescence (**b**,**e**,**h**), overlay (**c**,**f**,**i**). Representative images are shown.

**Figure 5 nanomaterials-13-01869-f005:**
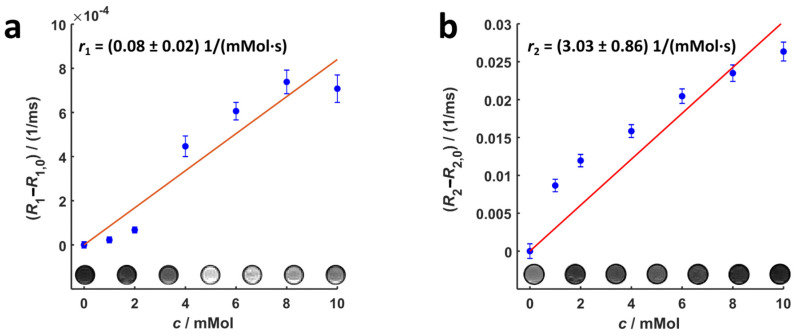
Longitudinal (**a**) and transversal (**b**) relaxation rates of Gd_2_O_3_-dex-RB nanoparticles vs. Gd_2_O_3_ concentration with a linear fit. The inset in (**a**,**b**) shows an exemplary magnetic resonance (MR) image for each concentration measured with an Inversion Recovery Spin Echo (IRSE) sequence (TR = 5000 ms, TI = 2500 ms, TE = 12 ms) and a Turbo Spin Echo (TSE) sequence (TR = 1500 ms, TE = 70 ms), respectively.

**Figure 6 nanomaterials-13-01869-f006:**
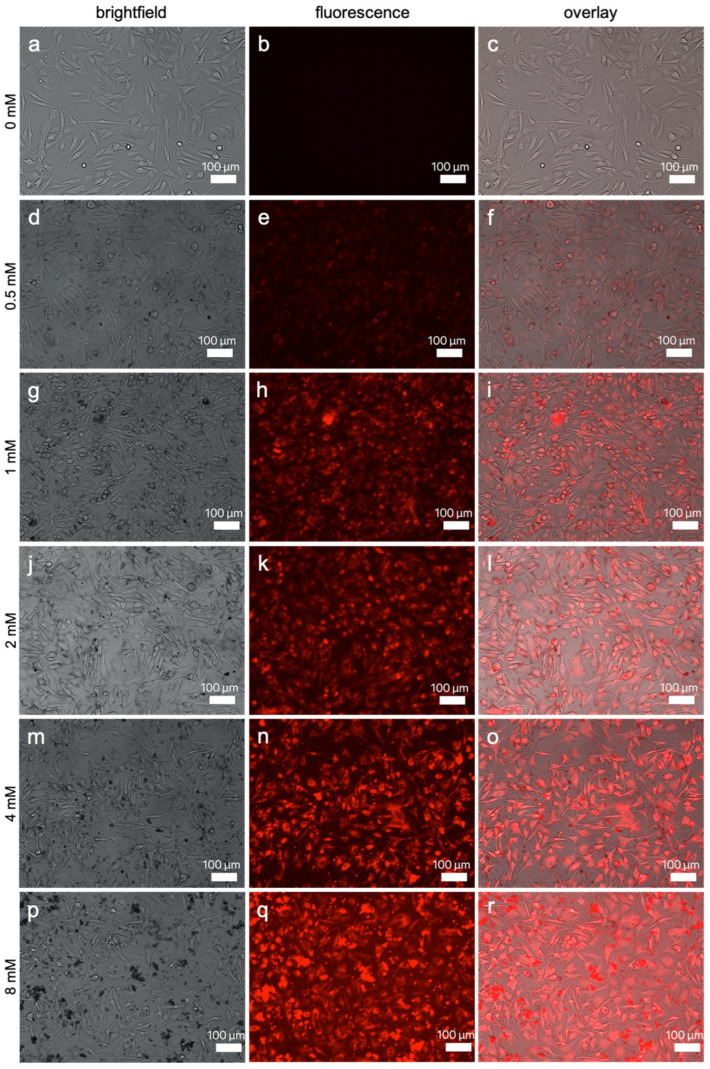
SK-MEL-28 incubated with increasing concentrations of Gd_2_O_3_-dex-RB nanoparticles for 24 h. Showing bright-field (**a**,**d**,**g**,**j**,**m**,**p**), fluorescence microscopy (**b**,**e**,**h**,**k**,**n**,**q**), and overlay (**c**,**f**,**i**,**l**,**o**,**r**). Cells were incubated with 0 mM (**a**–**c**), 0.5 mM (**d**–**f**), 1 mM (**g**–**i**), 2 mM (**j**–**l**), 4 mM (**m**–**o**), and 8 mM (**p**–**r**) Gd_2_O_3_-dex-RB nanoparticles for 2 h. Representative images are shown (*n* = 3).

**Figure 7 nanomaterials-13-01869-f007:**
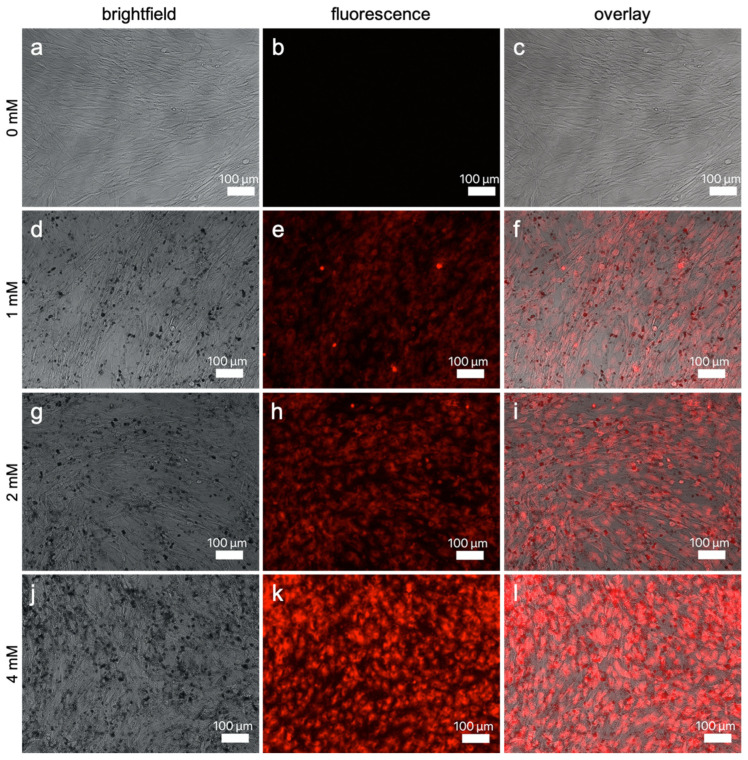
ASC labeled with ascending concentrations of Gd_2_O_3_-dex-RB nanoparticles. Microscopy using bright-field (**a**,**d**,**g**,**j**), fluorescence (**b**,**e**,**h**,**k**), and overlay (**c**,**f**,**i**,**l**). Cells were incubated with 0 mM (**a**–**c**), 1 mM (**d**–**f**), 2 mM (**g**–**i**), and 4 mM (**j**–**l**) Gd_2_O_3_-dex-RB nanoparticles for 24 h. Representative images are shown (*n* = 3).

**Figure 8 nanomaterials-13-01869-f008:**
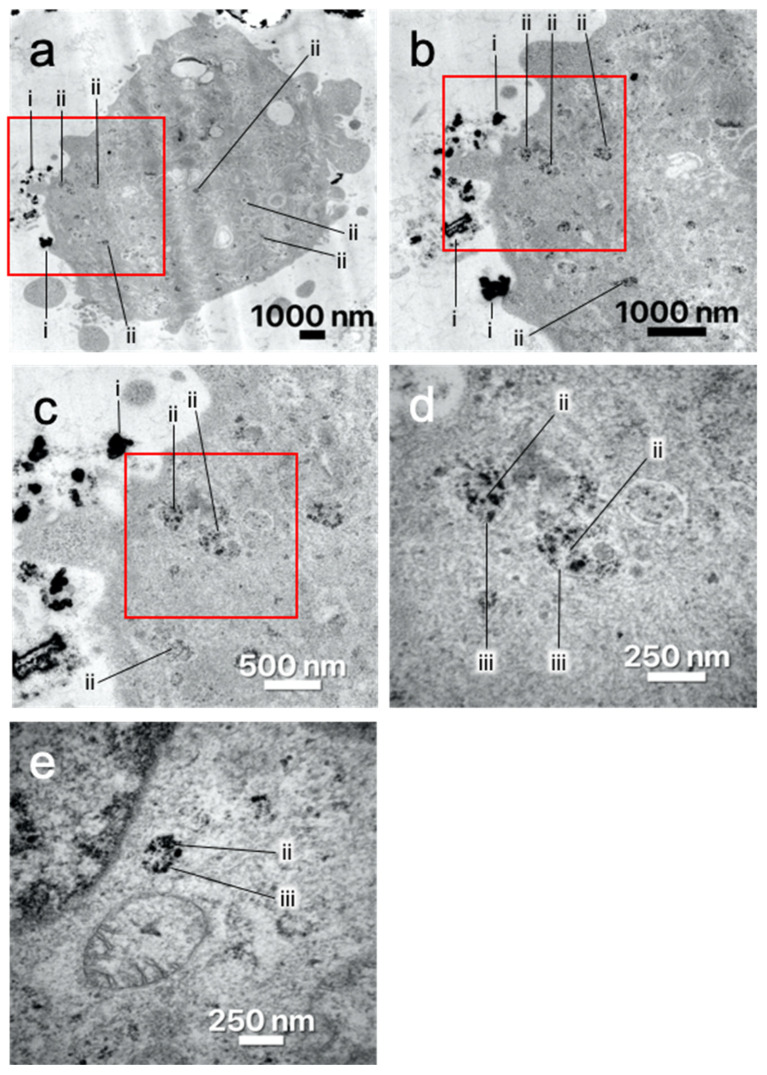
TEM imaging of SK-MEL-28 with external (i) and internalized (ii) Gd_2_O_3_-dex-RB nanoparticles. A thin membrane surrounding the internalized nanoparticles can be observed (iii). Cells were incubated with 4 mM Gd_2_O_3_-dex-RB nanoparticles for 24 h before fixation. Representative images are shown. (**b**–**e**) show magnifications of (**a**).

**Figure 9 nanomaterials-13-01869-f009:**
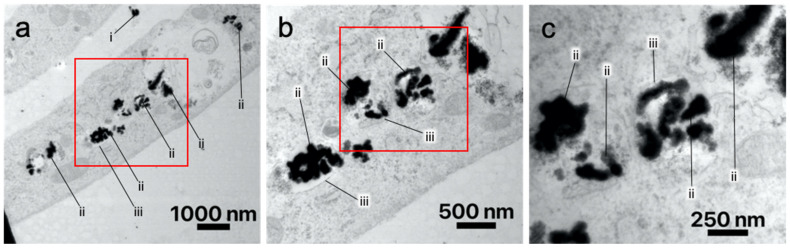
TEM imaging of ASC with external (i) and internalized (ii) Gd_2_O_3_-dex-RB nanoparticles. A thin membrane surrounding the internalized nanoparticles can be observed (iii). Cells were incubated with 1 mM Gd_2_O_3_-dex-RB nanoparticles for 24 h before fixation. Representative images are shown. (**b**,**c**) show magnifications of (**a**).

**Figure 10 nanomaterials-13-01869-f010:**
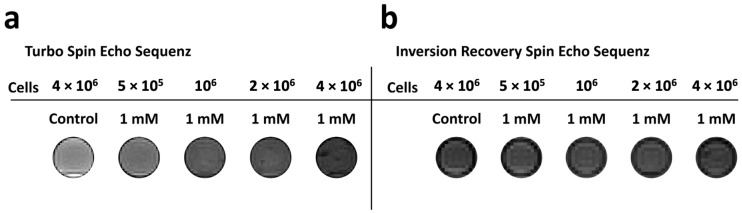
Exemplary MR images of SK-MEL-28 labeled with Gd_2_O_3_-dex-RB nanoparticles (1 mMol) measured with a TSE sequence (TR = 1500 ms, TE = 70 ms) (**a**) and an IRSE sequence (TR = 5000 ms, TI = 2500 ms, TE = 12 ms) (**b**), respectively.

**Figure 11 nanomaterials-13-01869-f011:**
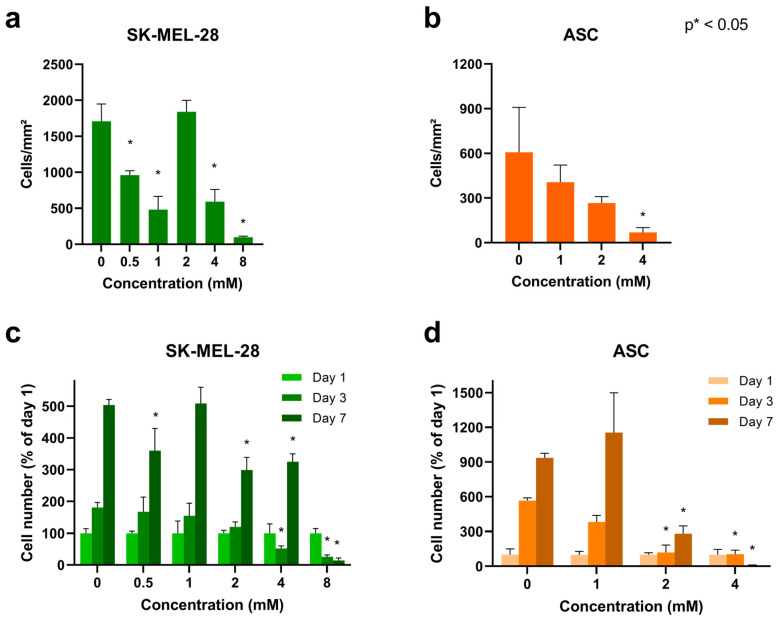
Cell proliferation of labeled SK-MEL-28 and ASC. Shown is the cell count on day 1 for SK-MEL-28 (**a**) and ASC (**b**) after incubating the cells with different concentrations of Gd_2_O_3_-dex-RB nanoparticles for 24 h, then trypsinizing and seeding in well plates. The cell number of SK-MEL-28 (**c**) and ASC (**d**) on day 3 and day 7 compared to day 1 is shown for different concentrations. The error bars represent the standard deviation of SK-MEL-28 using three technical replicates of each labeling concentration (*n* = 3) and of ASC using one well per donor (*n* = 3 independent donors).

**Table 1 nanomaterials-13-01869-t001:** Cell labeling concentrations in different assessments.

Assessment	Nanoparticle Labeling Concentrations (mM)	
	SK-MEL-28	ASC
TEM	0, 4	0, 1
Photometric Assessments: Spectrum Concentration series		
	0, 10	0, 4
	0, 0.5, 1, 2, 4, 8	0, 1, 2, 4
Fluorescence imaging	0, 0.5, 1, 2, 4, 8	0, 1, 2, 4
MRI	0, 1, 4	
Cell proliferation	0, 0.5, 1, 2, 4, 8	0, 1, 2, 4

**Table 2 nanomaterials-13-01869-t002:** Nanoparticle concentrations in different assessments.

Assessment	Sample	Concentrations (mM)
TEM	Gd_2_O_3_ nanoparticles Gd_2_O_3_-dex nanoparticles Gd_2_O_3_-dex-RB nanoparticles	1 1 1
Zeta potential	Gd_2_O_3_ nanoparticles Gd_2_O_3_-dex nanoparticles Gd_2_O_3_-dex-RB nanoparticles	10 10 10
Photometric Assessments:		
Spectrum	Gd_2_O_3_-dex-RB nanoparticles	10
Coating steps	Gd_2_O_3_ nanoparticles Gd_2_O_3_-dex nanoparticles Gd_2_O_3_-dex-RB nanoparticles	10 10 10
Concentration series	Gd_2_O_3_-dex-RB nanoparticles	0, 1, 2, 4, 6, 8, 10
Fluorescence imaging	Gd_2_O_3_ nanoparticles Gd_2_O_3_-dex nanoparticles Gd_2_O_3_-dex-RB nanoparticles	10 10 10
MRI	Gd_2_O_3_-dex-RB nanoparticles	0, 1, 2, 4, 6, 8, 10

## Data Availability

The data presented in this study are available on request from the corresponding author.

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
