# Peer review of "Dual Labeling of Primary Cells with Fluorescent Gadolinium Oxide Nanoparticles"

_nanomaterials, 2023, doi:10.3390/nano13121869_

Round 1

Reviewer 1 Report

The paper reports on the new preparation method of rose-bengal-dextran-coated gadolinium oxide nanoparticles and their use in dual-mode cell labeling. The subject of the paper falls within the scope of Nanomaterials journal and within the scope of the Special issue, too.

I have the following comments:

1. As it is stated in the Conclusions section, the authors proposed a new protocol for the synthesis of Gd2O3 nanoparticles coated with dextrane and functionalized with a rose bengal dye. Unfortunately, the protocol is not well documented and the properties of the nanoparticles are not well characterized. No information is presented concerning the hydrodynamic sizes of the particles at each stage of the synthesis. Particle size distributions are to be provided for the initial suspension of Gd2O3 particles in ultrapure water, for dextran-coated Gd2O3 nanoparticles, and, finally, for rose-bengal-dextran-coated Gd2O3 nanoparticles. Does the coating affect the hydrodynamic sizes of nanoparticles? Please comment on this. I would also like to emphasize that it is the hydrodynamic size of the particles that determine the means of their internalization. Please also consider the corresponding changes in the Discussion section when discussing the possible ways of internalization of functionalized Gd2O3 nanoparticles.

2. Next, no information is provided concerning the short-term and middle-term stability of the suspensions of Gd2O3 nanoparticles. Is it improved by functionalization, or no? What is the shelf life of these suspensions? Are there any signs of precipitation leading to the change of the actual NPs concentration?

3. Please provide the electron diffraction data (or X-ray diffraction data) to confirm the phase composition of the coated nanoparticles.

4. Please provide any available data on the solubility of Gd2O3. These data are of special importance because of the high toxicity of free gadolinium(III) ions.

5. Recently, several papers were published concerning the use of Gd-containing mixed oxides for MRI imaging and for cell labeling. A brief comparison of this approach and the approach proposed in the current paper could be useful for the readers.

Reviewer 2 Report

This manuscript describes interesting results on the use of dual labeling of stromal and melanoma cells. The authors provide a detailed description of all the procedures used for labeling. The obtained results show that the nanoparticles were successfully internalized without affecting cell viability. Overall, these results demonstrate that the Gb nanoparticles used are a feasible contrast agent for cell tracking. This is a well-written paper that can be published as is.

Author Response

Thank you for your supportive revision and valuable feedback on our manuscript, we appreciate the time you took to review our manuscript.

Reviewer 3 Report

The paper a new protocol and detailed investigation of cell labeling with fluorescent Gd2O3 nanoparticles. In my opinion, the research is made at a very high level, and very detailed and traceable descriptions of all the experiments. From the scientific contents, the paper raises no further questions.

The text has some minor italicization, paragraph style, and spacing errors.

Author Response

Thank you for your supportive revision and valuable feedback, we appreciate the time you took to review our manuscript and are grateful for you pointing out certain formal errors. We have carefully revised the manuscript again to correct these.

Reviewer 4 Report

Very accurate description of the new protocol to synthesize Gd oxude NPS which is trully one of the best MRI contrast agents. Fully suits the demands of Nanomaterials

Author Response

(The authors gave the same response as above.)

Round 2

Reviewer 1 Report

The authors have addressed most of my comments. The paper is now suitable for publication in Nanomaterials journal.